# Hierarchical Graph Matching Networks for Deep Graph Similarity Learning

## Abstract

While the celebrated graph neural networks yield effective representations for individual nodes of a graph, there has been relatively less success in extending to deep graph similarity learning. Recent work has considered either global-level graph-graph interactions or low-level node-node interactions, ignoring the rich cross-level interactions between parts of a graph and a whole graph. In this paper, we propose a Hierarchical Graph Matching Network (HGMN) for computing the graph similarity between any pair of graph-structured objects. Our model jointly learns graph representations and a graph matching metric function for computing graph similarity in an end-to-end fashion. The proposed HGMN model consists of a multi-perspective node-graph matching network for effectively learning cross-level interactions between parts of a graph and a whole graph, and a siamese graph neural network for learning global-level interactions between two graphs. Our comprehensive experiments demonstrate that our proposed HGMN consistently outperforms state-of-the-art graph matching network baselines for both classification and regression tasks.

## 1 Introduction

Learning a general similarity metric between arbitrary pairs of graph-structured objects is one of the key challenges in machine learning. Such learning problems often arise in a variety of applications, ranging from graph similar searching in graph-based database (Yan & Han, 2002), to Fewshot 3D Action Recognition (Guo et al., 2018), unknown malware detection (Wang et al., 2019), and promising selection in automatic theory proving (Wang et al., 2017), to name just a few.

Conceptually, classical exact (or inexact) graph matching techniques (Ullmann, 1976; Caetano et al., 2009; Bunke & Allermann, 1983; Riesen et al., 2010) provide a strong tool for learning graph similarity. However, these methods usually either require input graphs with similar sizes or consider mainly the graph structures for finding a correspondence between the nodes of different graphs without taking into account the node representations or features. In contrast, in this paper, we consider the graph matching problem of learning a mapping between a pair of graph inputs $(G^1, G^2) \in \mathcal{G} \times \mathcal{G}$ and the similarity score $y \in \mathcal{Y}$, based on a set of training triplet of structured input pairs and scalar output score $(G_1^1, G_1^2, y_1), ..., (G_n^1, G_n^2, y_n) \in \mathcal{G} \times \mathcal{G} \times \mathcal{Y}$ drawn from some fixed but unknown probability distribution.

Recent years have seen a surge of interests in graph neural networks (GNNs), which have been demonstrated to be a powerful class of models for learning node embeddings of graph-structured data (Bronstein et al., 2017). Various GNN models have since been developed for learning effective node representations for node classification (Li et al., 2016; Kipf & Welling, 2016; Hamilton et al., 2017; Veličković et al., 2017), or pooling the learned node embeddings into a graph vector for graph classification (Ying et al., 2018; Ma et al., 2019), or combining with variational auto-encoder to learn the graph distribution for graph generation (Simonovsky & Komodakis, 2018; Li et al., 2018; Samanta et al., 2018; You et al., 2018). However, there is relatively less study on learning graph similarity using GNNs.

To learn graph similarity, a simple yet straightforward way is to encode each graph as a vector and combine two vectors of each graph to make a decision. This approach is useful since graph-level embeddings contain important information of a pair of graphs. One obvious limitation of this approach lies in the fact of the ignorance of more fine-grained interactions among different level

embeddings of two graphs. Very recently, a few of attempts have been made to take into account low-level interactions either by considering the histogram information of node-wise similarity matrix of node embeddings (Bai et al., 2019) or improving the node embeddings of one graph by incorporating implicit attentive neighbors of another graphs through a soft attention (Li et al., 2019). However, there are two significant challenges making these graph matching models potentially ineffective: i) how to learn different-level granularity (global level and local level) of interactions between a pair of graphs; ii) how to effectively learn richer cross-level interactions between parts of a graph and a whole graph.

Inspired by these observations, in this paper, we propose a Hierarchical Graph Matching Network (HGMN) for computing the graph similarity between any pair of graph-structured objects. Our model jointly learns graph representations and a graph matching metric function for computing graph similarity in an end-to-end fashion. The proposed HGMN model consists of a novel multi-perspective node-graph matching network for effectively learning cross-level interactions between parts of a graph and a whole graph, and a siamese graph neural network for learning global-level interactions between two graphs. Our final small prediction networks consume these feature vectors from both cross-level and global-level interactions to perform either graph-graph classification or graph-graph regression tasks, respectively.

Recently proposed works only compute graph similarity by considering either graph-graph classification problem (with labels $Y = \{-1, 1\}$) (Li et al., 2019), or graph-graph regression problem (with similarity score $Y = [0, 1]$) (Bai et al., 2019). To demonstrate the effectiveness of our model, we systematically investigate the performance of our HGMN model compared with these recently proposed graph matching models on four datasets for both graph-graph classification and regression tasks. To bridge the gap of the lack of standard graph matching datasets, we also create one new dataset from a real application together with a previously released dataset by (Xu et al., 2017) for graph-graph classification task [1]. One important aspect is previous works did not consider the impact of the size of two input graphs, which often plays an important role in determining the performance of graph matching. Motivated by this observation, we have considered three different ranges of graph sizes from [3, 200], [20,200], and [50,200] in order to evaluate the robustness of each graph matching model.

We highlight our main contributions of this paper as follows:

- We propose a hierarchical graph matching network (HGMN) for computing the graph similarity between any pair of graph-structured objects. Our HGMN model jointly learns graph representations and a graph matching metric function for computing graph similarity in an end-to-end fashion.

- In particular, we propose a multi-perspective node-graph matching network for effectively capturing the cross-level interactions between a node embeddings of a graph and a corresponding attentive graph-level embedding of another graph.

- We systematically investigate different factors on the performance of all graph matching models such as the impact of different tasks (classification and regression) and the sizes of input graphs.

- Our comprehensive experiments demonstrate that our proposed HGMN consistently outperforms state-of-the-art graph matching network baselines for both classification and regression tasks. Compared with previous works, our proposed model HGMN is also more robust when the sizes of the two input graphs increase.

## 2   PROBLEM FORMULATION

In this section, we briefly introduce the problem formulation. Given a pair of graph inputs $(G^1, G^2)$, the aim of the graph matching problem we consider in this paper is to produce a graph similarity score $y = s(G^1, G^2) \in \mathcal{Y}$. The graph $G^1 = (V^1, E^1)$ is represented as a set of $N$ nodes $v_i \in V^1$ with a feature matrix $X^1 \in \mathcal{R}^{N \times d}$, edges $(v_i, v_j) \in E^1$ (binary or weighted) formulating an adjacency matrix $A^1 \in \mathcal{R}^{N \times N}$, and a degree matrix $D_{ii}^1 = \sum_j A_{ij}^1$. Similarly, the graph $G^2 = $

---

[1]We release these datasets via this link: `https://github.com/runningoat/hgmn_dataset`. Our codes will be released as well upon the acceptance of this paper.

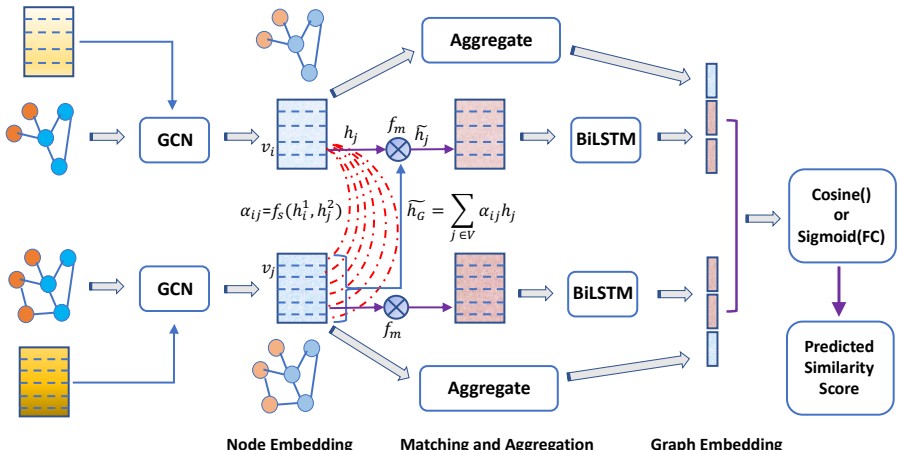

Figure 1: Overall Model Architecture of Hierarchical Graph Matching Networks (HGMN)

$(V^2, E^2)$ is represented as a set of $M$ nodes $v_i \in V^2$ with a feature matrix $X^2 \in \mathcal{R}^{M \times d}$, edges $(v_i, v_j) \in E^2$ (binary or weighted) formulating an adjacency matrix $A^2 \in \mathcal{R}^{M \times M}$, and a degree matrix $D_{ii}^2 = \sum_j A_{ij}^2$. Note that, when performing graph-graph classification task the scalar $y$ is the class labels $y = \{-1, 1\}$; when performing graph-graph regression task the scalar $y$ is the the measure of the similarity score $y \in [0, 1]$. We train a graph matching model based on a set of training triplet of structured input pairs and scalar output score $(G_1^1, G_1^2, y_1), ..., (G_n^1, G_n^2, Y_n) \in \mathcal{G} \times \mathcal{G} \times \mathcal{Y}$ drawn from some fixed but unknown probability distribution in real applications.

## 3 HIERARCHICAL GRAPH MATCHING NETWORKS ARCHITECTURE

In this section, we will introduce two key components of our HGMN architecture - Siamese Graph Neural Networks (SGNN) and Multi-Perspective Node-Graph Matching Networks (MPNGMN). We first discuss SGNN for learning the global-level interactions between two graphs and then outline MPNGMN for effectively learning the cross-level node-graph interactions between parts of one graph and one whole graph. Our overall model architecture for HGMN is shown in Fig. 1.

### 3.1 SGNN FOR GLOBAL-LEVEL INTERACTION LEARNING

The graph-level embeddings contain important information of a graph. Therefore, learning graph-level interactions between two graphs could be an important component for learning the graph similarity of two graphs. In order to capture the global-level interactions between two graphs, we employ SGNN which is based on Siamese Networks architecture (Bromley et al., 1994), which has achieved great success in many applications such as visual recognition (Bertinetto et al., 2016; Varior et al., 2016) and sentence similarity (He et al., 2015; Mueller & Thyagarajan, 2016). Independently, a similar idea using high-order siamese graph neural networks was presented for brain network analysis (Chaudhuri et al., 2019).

Our SGNN adapts popular Graph Convolution Networks (GCN) (Kipf & Welling, 2016) with siamese neural networks for simplicity. Other variants of graph neural networks such as Graph-SAGE (Hamilton et al., 2017) and Gated Graph Neural Networks (Li et al., 2016) can also be used. Our SGNN consists of three components: 1) node embedding layers; 2) graph-level embedding aggregation layers; 3) graph-graph matching and prediction layers.

**Node Embedding Layers.** We utilize three-layer GCN with the siamese networks to generate node embeddings $H^l = \{\mathbf{h}_i^l\}_{i=1}^{\{N,M\}} \in \mathcal{R}^{\{N,M\} \times d'}$ of both graphs $G^1$ and $G^2$,

$$H^l = f(X^l, A^l) = \text{ReLU}\Big(\bar{A}^l \, \text{ReLU}\Big(\bar{A}^l \, \text{ReLU}\Big(\bar{A}^l X^l W^{(0)}\Big) W^{(1)}\Big) W^{(2)}\Big), \; l = \{1, 2\}. \quad (1)$$

where $\bar{A}^l = (\widetilde{D}^l)^{-\frac{1}{2}} \widetilde{A}^l (\widetilde{D}^l)^{-\frac{1}{2}}$ is the normalized Laplacian matrix for $\widetilde{A}^l = A^l + I_{\{N,M\}}$ depending on the graph is $G^1$ or $G^2$, and $W^{(i)}, i = \{0, 1, 2\}$ are hidden weighted matrices for each layer. Note

that the twin networks share the parameters of GCN when training on the pair of graphs $(G^1, G^2)$. The number of GCN layers required depends on the real application graph data. To isolate the effect of overtuning, we choose the three layers after some initial experiments on validation sets.

**Graph-level Embedding Aggregation Layers.** After we compute the resulting node embeddings $H^l$ of each graph from GCN, we need to aggregate these node embeddings to formulate their corresponding graph-level embeddings of each graph.

$$\mathbf{h}_G^l = \text{Aggregate}\Big(\{\mathbf{h}_i^l\}_{i=1}^{\{N,M\}}\Big), \quad l = \{1, 2\}. \tag{2}$$

We employ different aggregation functions such as element-wise max pooling (Max), element-wise max pooling following a transformation by applying a fully connected layer on $H^i$ (FCMax), element-wise mean pooling (Avg), element-wise mean pooling following a transformation by applying a fully connected layer on $H^i$ (FCAvg), and a sophisticated aggregator based on LSTM architecture (Hochreiter & Schmidhuber, 1997a). Note that, among these aggregation functions, the LSTM aggregator is not permutation invariant on a set of node embeddings although LSTM may admit more expressive ability. We adapt LSTMs to operate on these node embeddings by simply applying the LSTMs to a random permutation of the node embeddings.

**Graph-Graph Matching and Prediction Layers.** After the graph-level embeddings $\mathbf{h}_G^1$ and $\mathbf{h}_G^2$ are computed for the graphs $G^1$ and $G^2$, we then use the resulting graph embeddings to compute the graph similarity score of $(G^1, G^2)$. Depending on the specific tasks, we have slightly different ways to calculate the final similarity score. For classification tasks, we simply compute the cosine similarity of two graph-level embeddings,

$$\widetilde{y} = s(G^1, G^2) = cosine(\mathbf{h}_G^1, \mathbf{h}_G^2) \tag{3}$$

where the similarity function $s$ could also be other similarity metric such as Euclidean similarity and dot-product similarity. We find that the cosine similarity function performs generally better across different datasets.

For regression tasks, we first concatenate the two aggregated graph embeddings to $[\mathbf{h}_G^1, \mathbf{h}_G^2]$ and then employ four standard fully connected layers to gradually project the vector of dimension $[\mathbf{h}_G^1, \mathbf{h}_G^2]$ down to a scalar of the dimension 1. Since the expected similar score $\widetilde{y}$ should be in range of [0,1], we perform sigmoid function to enforce the final score in this range. We therefore compute the similarity score for graph-graph regression task as following,

$$\widetilde{y} = s(G^1, G^2) = \text{sigmoid}\Big(\text{MLP}\Big([\mathbf{h}_G^1, \mathbf{h}_G^2]\Big)\Big). \tag{4}$$

For both tasks, we train the SGNN model using mean square error loss function to compare the computed similarity score $\widetilde{y}$ with the groud-truth similarity score $y$,

$$\mathcal{L} = \frac{1}{n} \sum_{i=1}^{n} \Big(\widetilde{y} - y\Big)^2. \tag{5}$$

### 3.2 MPNGMN for Cross-level node-graph interaction learning

Although global-level interaction learning could capture the important structural and feature information of two graphs to some extent, it is not enough to capture all important information of two graphs since they ignore other low-level interactions between parts of two graphs. In particular, existing works have considered either global-level graph-graph interactions or low-level node-node interactions, ignoring the rich cross-level interactions between parts of a graph and a whole graph. Inspired by these observations, we propose a novel multi-perspective node-graph matching network to effectively learn the cross-level interaction features. Our MPNGMN model consists of four parts: 1) node embedding layers; 2) node-graph matching layers; 3) aggregation layers; and 4) prediction layers, as shown in Fig. 1. We will illustrate each part in details as follows.

**Node Embedding Layers:** Similar as described in Sec. 3.1, we choose to employ the three-layer GCN to generate node embeddings $H^1 = \{\mathbf{h}_i^1\}_{i=1}^{N} \in \mathcal{R}^{N \times d'}$ and $H^2 = \{\mathbf{h}_i^2\}_{i=1}^{M} \in \mathcal{R}^{M \times d'}$ for graphs $G^1$ and $G^2$. Conceptually, the node embedding layers of MPNGMN (graph encoder) could be chosen to be an independent GCN or a shared GCN with SGNN. As shown in Fig. 1, our MPNGMN shares the same graph encoder with SGNN due to two reasons: i) the shared GCN parameters reduce the number of parameters by half, which helps mitigate possible overfitting; ii) the shared

GCN maintains the consistency of resulting node embeddings for both MPNGMN and SGNN, potentially leading to more aligned global-level interaction and cross-level interaction features. After the node embeddings $H^1$ and $H^2$ have been computed, they will be fed into the following node-graph matching layers.

**Node-Graph Matching Layers:** The node-graph matching layer is the key part of our MPNGMN, which can effectively learn the cross-level interactions between parts of a graph and a whole graph. There are generally two steps for this layer: i) calculate the graph-level embedding of a graph; ii) compare the node embeddings of a graph with the associated graph-level embeddings of a whole graph and then produce a similarity feature vector. A simple way to obtain the graph-level embedding of a graph is to perform element-wise mean pooling or max pooling. However, it does not consider any information from the node embedding that the resulting graph-level embedding will compare with later. To build more tight interactions between the two, we calculate the cross-graph attention coefficients between the node $v_i \in \mathcal{V}^1$ in graph $G^1$ and all other nodes $v_j \in \mathcal{V}^2$ in graph $G^2$. Similarly, we calculate the cross-graph attention coefficients between the node $v_i \in \mathcal{V}^2$ in graph $G^2$ and all other nodes $v_j \in \mathcal{V}^1$ in graph $G^1$. These two cross-graph attention coefficients can be computed independently,

$$\alpha_{i,j} = f_s(\mathbf{h}_i^1, \mathbf{h}_j^2),\ j \in \mathcal{V}^2 \quad \text{and} \quad \beta_{i,j} = f_s(\mathbf{h}_i^2, \mathbf{h}_j^1),\ j \in \mathcal{V}^1, \tag{6}$$

where $f_s$ is the attention function for computing the similarity score. For simplicity, we use cosine function in our experiments but other similarity metrics can be adopted as well. Then we compute the attentive graph-level embeddings $\widetilde{\mathbf{h}}_G^1$ or $\widetilde{\mathbf{h}}_G^2 \in \mathcal{R}^{d'}$ using weighted average of node embeddings of the other graph,

$$\widetilde{\mathbf{h}}_G^2 = \sum_{j \in \mathcal{V}^2} \alpha_{i,j} \mathbf{h}_j^2 \quad \text{and} \quad \widetilde{\mathbf{h}}_G^1 = \sum_{j \in \mathcal{V}^1} \beta_{i,j} \mathbf{h}_j^1. \tag{7}$$

Next, we define our multi-perspective matching function $f_m$ to compute the similarity feature vector by comparing two vectors as follows,

$$\widetilde{\mathbf{h}}(i) = f_m(\mathbf{x}_1, \mathbf{x}_2, \mathbf{w}_i) = f_m(\mathbf{x}_1 \odot \mathbf{w}_i, \mathbf{x}_2 \odot \mathbf{w}_i),\ i = 1, \ldots, \widetilde{d} \tag{8}$$

where $\widetilde{\mathbf{h}} \in \mathcal{R}^{\widetilde{d}}$ is a $\widetilde{d}$-dimension similarity feature vector, and $W_m = \{\mathbf{w}_i\}_{i=1}^{\widetilde{d}} \in \mathcal{R}^{d' \times \widetilde{d}}$ is a trainable weight matrix and each $\mathbf{w}_i$ represents a perspective with total $\widetilde{d}$ number of perspectives. Notably, $f_m$ could be any similarity function and we use cosine similarity metric in our experiments. It is worth noting that the proposed multi-perspective matching function essentially shares similar spirit with multi-head attention (Vaswani et al., 2017), with the difference that multi-head attention uses $\widetilde{d}$ number of weighted matrices instead of vectors.

Therefore, we can utilize our defined multi-perspective matching function $f_m$ to compare the $j$-th node embeddings of a graph with the corresponding attentive graph-level embeddings to capture the cross-level node-graph interactions. The resulting similarity feature vectors $\widetilde{\mathbf{h}}_j^1$ or $\widetilde{\mathbf{h}}_j^2 \in \mathcal{R}^{\widetilde{d}}$ (w.r.t the node $v_j$ in either graph $G^1$ or $G^2$) can thus be computed by

$$\widetilde{\mathbf{h}}_j^1 = f_m(\mathbf{h}_j^1, \widetilde{\mathbf{h}}_G^2, W_m),\ j \in \mathcal{V}_1 \quad \text{and} \quad \widetilde{\mathbf{h}}_j^2 = f_m(\mathbf{h}_j^2, \widetilde{\mathbf{h}}_G^1, W_m),\ j \in \mathcal{V}_2 \tag{9}$$

After the node-graph matching layers, these newly produced interaction feature matrices $\widetilde{H}^1 = \{\widetilde{\mathbf{h}}_i^1\}_{i=1}^N \in \mathcal{R}^{N \times \widetilde{d}}$ and $\widetilde{H}^2 = \{\widetilde{\mathbf{h}}_i^2\}_{i=1}^M \in \mathcal{R}^{M \times \widetilde{d}}$ for graphs $G^1$ and $G^2$, are ready to feed them into the aggregation layers.

**Aggregation Layers:** To aggregate these cross-level interaction feature matrix from the node-graph matching layer, we employ the BiLSTM (Hochreiter & Schmidhuber, 1997b) model to aggregate the unordered feature embeddings,

$$\widetilde{\mathbf{h}}_G^l = \text{BiLSTM}\left(\{\widetilde{\mathbf{h}}_j^l\}_{j=1}^{\{N,M\}}\right), \quad l = \{1, 2\}. \tag{10}$$

where $\widetilde{\mathbf{h}}_G^l \in \mathcal{R}^{2\widetilde{d}}$ concatenate the last hidden vectors of two directions as the aggregated graph embedding for each graph $G^1$ and $G^2$. Note that other commutative aggregators such as max, average, or attention based aggregation (Veličković et al., 2017) can also be used. However, our extensive experiments showed that BiLSTM aggregator achieved consistent better performance over

other aggregators. Similar LSTM-type aggregator has also been exploited in the previous works (Hamilton et al., 2017; Zhang et al., 2019).

**Prediction Layers:** After the aggregated graph embeddings $\widetilde{\mathbf{h}}_G^1$ and $\widetilde{\mathbf{h}}_G^2$ are obtained, we then use these two embeddings to compute the similarity score of $(G^1, G^2)$. As discussed in Sec.3.1 for graph-graph matching and prediction layers, we use the same prediction layers to predict the similarity score. We also use the same mean square error loss function for the model training. In this way, we can also easily compare the performance difference between SGNN and MPNGMN.

### 3.3 DISCUSSIONS ON HGMN MODEL

Our model jointly learns graph representations and a graph matching metric function for computing graph similarity in an end-to-end fashion. Our HGMN model combines the advantages of both SGNN and MPNGMN to capture both global-level graph-graph interaction features and novel cross-level node-graph interaction features between two graphs. Therefore, for final prediction layers of HGMN, we have total six aggregated graph embedding vectors where two of them are $\mathbf{h}_G^1$ and $\mathbf{h}_G^2$ from SGNN, and another four are $\widetilde{\mathbf{h}}_G^1$ and $\widetilde{\mathbf{h}}_G^2$ from MPNGMN.

The computation complexity of SGNN is $O((|E^1| + |E^2|)dd')$, where the most dominant computation is sparse matrix-matrix operations in equation 1. Similarly, the computational complexity of MPNGMN is $O(NMd + (N + M)d' + (N + M)dd')$, where the most computationally extensive operations are in equations 7, 8, and 9. Compared to recently proposed works in (Bai et al., 2019; Li et al., 2019), the computational complexity of them are comparable.

## 4 EXPERIMENTS

In this section, we systematically investigate the performance of our HGMN model compared with other recently proposed graph matching models on four datasets for both classification and regression tasks.

Table 1: Summary statistics of datasets for both classification & regression tasks.

| Tasks | Datasets | Sub-datasets | # of Graphs | # of Functions | AVG # of Nodes | AVG # of Edges | AVG # of AVG Degrees | Init Feature Dimensions |
|---|---|---|---|---|---|---|---|---|
| classif-ication | **FFmpeg** | [3, 200] | 83008 | 10376 | 18.83 | 27.02 | 2.59 | |
| | | [20, 200] | 31696 | 7668 | 51.02 | 75.88 | 2.94 | 6 |
| | | [50, 200] | 10824 | 3178 | 90.93 | 136.83 | 3.00 | |
| | **OpenSSL** | [3, 200] | 73953 | 4249 | 15.73 | 21.97 | 2.44 | |
| | | [20, 200] | 15800 | 1073 | 44.89 | 67.15 | 2.95 | 6 |
| | | [50, 200] | 4308 | 338 | 83.68 | /127.75 | 3.04 | |
| regre-ssion | **AIDS700** | - | 700 | - | 8.90 | 8.80 | 1.96) | 29 |
| | **LINUX1000** | - | 1000 | - | 7.58 | 6.94 | 1.81 | 1 |

### 4.1 DATASETS, EXPERIMENTS SETTINGS, AND BASELINES

#### 4.1.1 DATASETS

**Classification datasets**: we evaluate our model on the problem of detecting similarity between two binary functions, which is the heart of many binary security problems, such as software plagiarism, malware detection, and vulnerability search (Feng et al., 2016; Xu et al., 2017; Ding et al., 2019). In particular, two binary functions that are compiled from the same source code but under different settings (architectures, compilers, optimization levels, etc) are semantically similar to each other. To learn similarity from binary functions, we represent those binaries with control flow graphs, in which the graph nodes represent the basic blocks (a basic block is a sequence of instructions without jumps) and edges represent control flow paths between these basic blocks.

Thus, detecting similarity between two binary functions can be cast as the problem of learning the similarity score $s(G^1, G^2)$ between two control flow graphs $G^1$ and $G^2$, where $s(G^1, G^2) = +1$ indicates $G^1$ and $G^2$ are similar; otherwise $s(G^1, G^2) = -1$ indicates dissimilar. We prepare

two benchmark datasets generated from two pieces of popular open-source software: **FFmpeg** and **OpenSSL**, with statistics shown in Table 1. For each graph in **FFmpeg** and **OpenSSL**, we initialize every node with 6 block-level numeric features. More details about the dataset generation and node features can be found in Appendix A.1.1 and Table 7.

Existing graph matching works do not consider the impact of the sizes of graphs on performance. However, we find that the larger the graph size is, the worse the performance is. Therefore, it is important to evaluate the robustness of any graph matching networks in this setting. We thus further split these two datasets into three sub-datasets according to the size range of graph pairs.

**Regression datasets**: we evaluate our model on learning the graph edit distance (GED) (Zeng et al., 2009; Gao et al., 2010; Riesen, 2015), which measures the structural similarity between two graphs. Formally, GED is defined as the cost of the least expensive sequence of edit operations that transform one graph into another, where an edit operation can be an insertion or a deletion of a node or an edge.

We evaluate our model on two benchmark datasets **AIDS700** and **LINUX1000** [2]. The statistic for the datasets is shown in Table 1, and more details can be found in Appendix A.1.2 and Table 7.

### 4.1.2 EXPERIMENTAL SETUP

**Model Settings**. For SGNN, we use 3 GCN layers in node embedding layer and each of the GCNs has an output dimension of 100. We use ReLU as the activation function along with a dropout layer after each GCN layer with dropout rate being 0.1. In the graph-level embedding aggregation layer of SGNN, we can employ different aggregation functions (i,e., Max, FCMax, Avg, FCAvg, BiLSTM, etc.) as stated previously in Section 3.1. For MPNGMN, we exploited different aggregation functions similar to SGNN and we found that BiLSTM aggregator consistently performs better than other aggregation functions (see appendix A.4). Thus, for MPNGMN, we always use BiLSTM as our default aggregation function and we make its hidden size equal to the dimension of node embeddings. For MPNGMN, we set the number of perspectives $\widetilde{d}$ to 100, and use another aggregation function BiLSTM to aggregate the output of node-graph matching layer. For each graph, we concatenate the last hidden vector of two directions of BiLSTM, which results in a 200 dimensions vector as the graph embeddings.

**Implementation Details**. We implement our model using PyTorch 1.1 (Paszke et al., 2017), and train the model using the Adam optimizer (Kingma & Ba, 2014). The learning rate is set to 0.5e-3 for classification tasks and 5e-3 for regression tasks. For classification tasks, we split each dataset into three disjoint subsets of *binary functions* for training/validation/testing. We train our model by running 100 epochs. At each epoch, we build the pairwise training data as follows. For each graph $G$ in training subset, we obtain one positive pair $\{(G, G^{pos}), +1\}$ and a corresponding negative pair $\{(G, G^{neg}), -1\}$, where $G^{pos}$ is randomly selected from all control flow graphs that compiled from the same source function as $G$, and $G^{neg}$ is selected from other graphs. By default, for each mini-batch in one epoch, we train our model with 5 positive and 5 negative pairs. In regression tasks, we first split graphs of each dataset into training, validation, and testing set, and then build the pairwise training/validation/testing data as previous work Bai et al. (2019). We train our model by 10000 iterations with a mini-batch of 128 graph pairs. Each pair is a tuple of $\{(G^1, G^2), s\}$, where $s$ is the ground-truth GED between $G^1$ and $G^2$. Noted that all experiments are conducted on a computer equipped with 2 Intel Xeon 2.2GHz CPU, 256 GB memory and one NVIDIA GTX 1080 Ti GPU.

**Baselines.** We compared our HGMN against the following baselines: i) *SimGNN* (Bai et al. (2019)): SimGNN uses GCN to update node features and aggregates them using an attention mechanism. The final pair representation consists of 2 components: One from the interaction between aggregated pair graph features and the other from a pairwise node comparison. ii) *GMN* (Li et al. (2019)): This method updates node features according to not only current states and messages aggregated from neighborhoods but also information of attentive neighborhoods using cross-graph attention. After updating node features, it aggregates node features in a way similar to that in Gated Graph Neural Network (Li et al. (2016)) to get graph embedding. We have two variants of HGMN: HGMN(FCMax) stands for HGMN model with SGNN(FCMax) and HGMN(BiLSTM) stands for HGMN model with SGNN(BiLSTM).

---

[2] These two datasets are released by Bai et al. (2019) and publicly accessible.

Note that, we report the mean and standard deviation of the experimental results of both baseline and our models by repeating the experiments five times.

## 4.2 COMPARISON ON GRAPH-GRAPH CLASSIFICATION TASK

Table 2: Summary of classification results in terms of AUC scores (%).

| Model | FFmpeg | | | OpenSSL | | |
|---|---|---|---|---|---|---|
| | [3, 200] | [20, 200] | [50, 200] | [3, 200] | [20, 200] | [50, 200] |
| SimGNN | 95.38±0.76 | 94.31±1.01 | 93.45±0.54 | 95.96±0.31 | 93.58±0.82 | 94.25±0.85 |
| GMN | 94.15±0.62 | 95.92±1.38 | 94.76±0.45 | 96.43±0.61 | 93.03±3.81 | 93.91±1.65 |
| SGNN (Max) | 93.92±0.07 | 93.82±0.28 | 85.15±1.39 | 91.07±0.10 | 88.94±0.47 | 82.10±0.51 |
| MPNGMN | 97.73±0.11 | 98.29±0.21 | 96.81±0.96 | 96.56±0.12 | **97.60±0.29** | 92.89±1.31 |
| HGMN (FCMax) | **98.07±0.06** | **98.29±0.10** | **97.83±0.11** | 96.87±0.24 | 97.59±0.24 | 95.58±1.13 |
| HGMN (BiLSTM) | 97.56±0.38 | 98.12±0.04 | 97.16±0.53 | **96.90±0.10** | 97.31±1.07 | **95.87±0.88** |

For the classification task of detecting whether two binary functions are similar or not, we measure the *Area Under the ROC Curve (AUC)* (Bradley, 1997) of different models for classifying graph pairs of the same test set, and summarize the results in Table 2.

The results show that our models clearly achieve state-of-the-art performance on all 6 sub-datasets for both **FFmpeg** and **OpenSSL** datasets. Both MPNGMN and HGMN models show better and more robust performance than the SimGNN and GMN baselines, particularly when the graph size of the two graphs increases. Compared with the SGNN (Max), our models (both MPNGMN and HGMN models) significantly outperform it, demonstrating the benefits of multi-perspective node-graph matching mechanism that captures the cross-level interactions between node embeddings of a graph and graph-level embeddings of another graph. More experiments compared with SGNN models using other aggregation functions can be found in Appendix A.3.

## 4.3 COMPARISON ON GRAPH-GRAPH REGRESSION TASK

Table 3: Summary of regression results on **AIDS700** and **LINUX1000**.

| Datasets | Model | $mse$ $(10^{-3})$ | $\rho$ | $\tau$ | p@10 | p@20 |
|---|---|---|---|---|---|---|
| **AIDS700** | SimGNN | 1.376±0.066 | 0.824±0.009 | 0.665±0.011 | 0.400±0.023 | 0.489±0.024 |
| | GMN | 4.610±0.365 | 0.672±0.036 | 0.497±0.032 | 0.200±0.018 | 0.263±0.018 |
| | SGNN (Max) | 2.822±0.149 | 0.765±0.005 | 0.588±0.004 | 0.289±0.016 | 0.373±0.012 |
| | MPNGMN | 1.191±0.048 | 0.904±0.003 | 0.749±0.005 | **0.465±0.011** | 0.538±0.007 |
| | HGMN (FCMax) | 1.205±0.039 | 0.904±0.002 | 0.749±0.003 | 0.457±0.014 | 0.532±0.016 |
| | HGMN (BiLSTM) | **1.169±0.036** | **0.905±0.002** | **0.751±0.003** | 0.456±0.019 | **0.539±0.018** |
| **LINUX 1000** | SimGNN | 2.479±1.038 | 0.912±0.031 | 0.791±0.046 | 0.635±0.328 | 0.650±0.283 |
| | GMN | 2.571±0.519 | 0.906±0.023 | 0.763±0.035 | 0.888±0.036 | 0.856±0.040 |
| | SGNN (Max) | 11.832±0.698 | 0.566±0.022 | 0.404±0.017 | 0.226±0.106 | 0.492±0.190 |
| | MPNGMN | 1.561±0.020 | 0.945±0.002 | 0.814±0.003 | 0.743±0.085 | 0.741±0.086 |
| | HGMN (FCMax) | 1.575±0.627 | 0.946±0.019 | 0.817±0.034 | 0.807±0.117 | 0.784±0.108 |
| | HGMN (BiLSTM) | **0.439±0.143** | **0.985±0.005** | **0.919±0.016** | **0.955±0.011** | **0.943±0.014** |

For the regression task of computing the graph edit distance between two graphs, we evaluate the models using Mean Square Error ($mse$), Spearmans Rank Correlation Coefficient ($\rho$) (Spearman, 1904), Kendalls Rank Correlation Coefficient ($\tau$) (Kendall, 1938), and precision at k (p@k). All results of both **AIDS700** and **LINUX1000** datasets are summarized in Table 3. In terms of all evaluation metrics, our models consistently outperform both SimGNN and GMN baseline models by a significant margin on both **AIDS700** and **LINUX1000** datasets. On the other hand, compared with SGNN (Max), our models achieve much better performance (see Appendix A.3 for more experiments compared with other SGNN models). The results highlight the importance of our multi-perspective node-graph matching mechanism which could effectively capture cross-level node-graph interactions between parts of a graph and a whole graph.

Table 4: Classification results of Multi-Perspectives versus Multi-Heads in terms of AUC scores(%).

| Model | FFmpeg | | | OpenSSL | | |
|---|---|---|---|---|---|---|
| | [3, 200] | [20, 200] | [50, 200] | [3, 200] | [20, 200] | [50, 200] |
| Multi-Perspectives ($\tilde{d} = 100$) | 97.73±0.11 | 98.29±0.21 | 96.81±0.96 | 96.56±0.12 | 97.60±0.29 | 92.89±1.31 |
| Multi-Heads ($K = 6$) | 91.18±5.91 | 77.49±5.21 | 68.15±6.97 | 92.81±5.21 | 85.43±5.76 | 56.87±7.53 |

## 4.4 FURTHER STUDY ON THE IMPACT OF DIFFERENT ATTENTION FUNCTIONS

We perform a further study on the impact of different attention functions for our proposed MP-NGMN model. In particular, as we discussed in Sec. 3.2, the proposed multi-perspective matching function shares similar spirits with multi-head attention (Vaswani et al., 2017). Therefore, it is interesting to compare both attention functions in terms of AUC scores for graph-graph classification tasks. Interestingly, our proposed multi-perspective attention mechanism consistently outperforms these results of multi-head attention mechanism by quite a large margin. We suspect that our proposed multi-perspective attention uses vectors attention weights which may significantly reduce the potential overfitting. We also performed a study on the impact of the number of the perspectives on the performance and our model is not sensitive with this hyperparameter (see the appendix A.5).

## 4.5 FURTHER STUDY ON THE IMPACT OF DIFFERENT GNNS

Table 5: Classification results of different GNNs in terms of AUC scores (%).

| Model | FFmpeg | | | OpenSSL | | |
|---|---|---|---|---|---|---|
| | [3, 200] | [20, 200] | [50, 200] | [3, 200] | [20, 200] | [50, 200] |
| MPNGMN-GCN (Our) | 97.73±0.11 | 98.29±0.21 | 96.81±0.96 | 96.56±0.12 | 97.60±0.29 | 92.89±1.31 |
| MPNGMN-GraphSAGE | 97.31±0.56 | 98.21±0.13 | 97.88±0.15 | 96.13±0.30 | 97.30±0.72 | 93.66±3.87 |
| MPNGMN-GIN | 97.97±0.08 | 98.06±0.22 | 94.66±4.01 | 96.98±0.20 | 97.42±0.48 | 92.29±2.23 |
| MPNGMN-GGNN | **98.42±0.41** | **99.77±0.07** | **97.93±1.18** | **99.35±0.06** | **98.51±1.04** | **94.17±7.74** |

Table 6: Regression results of different GNNs on **AIDS700** and **LINUX1000**.

| Datasets | Model | $mse$ ($10^{-3}$) | $\rho$ | $\tau$ | p@10 | p@20 |
|---|---|---|---|---|---|---|
| **AIDS 700** | MPNGMN-GCN (Our) | **1.191±0.048** | **0.904±0.003** | **0.749±0.005** | **0.465±0.011** | **0.538±0.007** |
| | MPNGMN-(GraphSAGE) | 1.275±0.054 | 0.901±0.006 | 0.745±0.008 | 0.448±0.016 | 0.533±0.014 |
| | MPNGMN-(GIN) | 1.367±0.085 | 0.889±0.008 | 0.729±0.010 | 0.400±0.022 | 0.492±0.021 |
| | MPNGMN-(GGNN) | 1.870±0.082 | 0.871±0.004 | 0.706±0.005 | 0.388±0.015 | 0.457±0.017 |
| **LINUX 1000** | MPNGMN-GCN (Our) | 1.561±0.020 | 0.945±0.002 | 0.814±0.003 | 0.743±0.085 | 0.741±0.086 |
| | MPNGMN-GraphSAGE | 2.784±0.705 | 0.915±0.019 | 0.767±0.028 | 0.682±0.183 | 0.693±0.167 |
| | MPNGMN-GIN | **1.126±0.164** | **0.963±0.006** | **0.858±0.015** | **0.792±0.068** | **0.821±0.035** |
| | MPNGMN-GGNN | 2.068±0.991 | 0.938±0.028 | 0.815±0.055 | 0.628±0.189 | 0.654±0.176 |

We finally investigate the impact of different GNNs adopted by node embedding layers of our MP-NGMN model for both classification and regression tasks. Following the same settings of our previous experiments, we only replace GCN with three variants: GraphSAGE (Hamilton et al., 2017), GIN (Xu et al., 2018a), and GGNN (Li et al., 2016), whose output dimensions are kept the same with GCN (i.e, 100) in our experiments. Note that, we do not fine-tune any hyper-parameter of the three GNN models, and their default hyper-parameters of these three GNNs are listed in Appendix A.2.2.

Table 5 and Table 6 present the results of GCN versus GraphSAGE/GIN/GGNN in MPNGMN for the classification and regression tasks, respectively. For all datasets of classification and regression tasks, the performance of different GNNs is quite similar. It indicates that our model is not sensitive to the choice of GNN models in node embedding layers. Moreover, we can see from Table 5 that MPNGMN models using GGNN perform even better than our default MPNGMN using GCN on both **FFmpeg** and **OpenSSL** datasets for the classification task. It is also observed from Table 6 that MPNGMN models using GIN also outperform our default model using GCN on **LINUX1000** dataset for the regression task. These observations show that our model can be further improved by adopting more advanced GNN models or choosing the most appropriate GNN models according to different application tasks.

## 5 RELATED WORKS

**Graph Neural Networks.** Recently graph neural networks have been proven to be extremely effective and achieved promising results on various graph-structured based prediction tasks (Gao et al., 2019; Chen et al., 2019a). The main goal of graph neural networks is to learn node-level representations or (sub)graph-level representations for graph-structured data. There is a large body of GNN models (Scarselli et al., 2008; Li et al., 2016; Kipf & Welling, 2016; Hamilton et al., 2017; Veličković et al., 2017; Xu et al., 2018a) that have been proposed to learn node representations. With the learned node representations, various tasks on graphs can be performed such as node classification and link prediction (Veličković et al., 2017; Zhang & Chen, 2018). In addition to learning node representation, some studies try to extend pooling operations to GNNs (Ying et al., 2018; Gao & Ji, 2019; Lee et al., 2019; Ma et al., 2019). These pooling operations are expected to learn scaled-down graph representations from node representations, and can be trained in an end-to-end fashion. Recent works also exploit extending sequence-to-sequence model using bidirectional GNN for developing graph-to-sequence models in order to cope with graph inputs and show promising performance improvement (Xu et al., 2018b;c; Chen et al., 2019b) in various natural language processing tasks.

**Conventional Graph Matching.** In general, graph matching can be categorized into *exact graph matching* and *error-tolerant graph matching*. Exact graph matching aims to find a strict correspondence between two (in large parts) identical graphs being matched, while error-tolerant graph matching allows matching between completely nonidentical graphs (Riesen, 2015). In real-world applications, the constraint of exact graph matching is too rigid, and thus a large number of work has been proposed to solve the error-tolerant graph matching problem, which is usually quantified by a specific similarity metric. In fact, the matching similarity metrics can be defined by some measure of structure similarity like Graph Edit Distance (GED) (Gao et al., 2010), Maximum Common Subgraph (MCS) (Bunke, 1997), or even more coarse binary similarity, according to different application backgrounds. For GED and MCS, both of them are well-studies NP-hard problems (Bunke, 1997; McGregor, 1982), and thus suffer from exponential computational complexity and huge memory requirements for exact solutions in practice (Zeng et al., 2009; Blumenthal & Gamper, 2018).

**Graph Similarity Computation and Graph Matching Networks.** A popular line of research of graph matching focuses on developing approximations for graph similarity computations, in which most of them focus on improvements for better efficiency in computation (Gao et al., 2010; Zeng et al., 2009; Riesen, 2015; Wu et al., 2019; Yoshida et al., 2019). However, our solution is a learnable model based on GNN to approximate graph similarity in terms of GED and binary similarity for pairwise graph-based data.

The closely relevant work to our solution are two GNN based models: GMN (Li et al., 2019) and SimGNN (Bai et al., 2019). GMN directly updates the node representations of one graph by adding artificial attention-based connections for another graph. SimGNN considers the graph-level representation similarity as well as the histogram features from a pairwise node-level comparison to learn the graph similarity. However, these two models fail to capture different perspectives of graph-structured data between the pairs of graphs.

## 6 CONCLUSION AND FUTURE WORK

In this paper, we presented a novel Hierarchical Graph Matching Network (HGMN) for computing the graph similarity between any pair of graph-structured objects. Our model jointly learned graph embeddings and a data-driven graph matching metric for computing graph similarity in an end-to-end fashion. We further proposed a new multi-perspective node-graph matching network for effectively learning cross-level interactions between two graphs beyond low-level node-node and global-level graph-graph interactions. Our extensive experimental results correlated the superior performance compared with state-of-the-art baselines on both graph-graph classification and regression tasks.

One interesting future direction is to adapt our proposed HGMN model for solving different real-world applications such as unknown malware detection, text matching and entailment, and knowledge graph question answering.

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

## A  APPENDIX

### A.1  DATASETS

#### A.1.1  CLASSIFICATION DATASETS

It is noted that one source code function, after compiled with different settings (architectures, compilers, optimization levels, etc), can generate various binary functions with different control flow graphs.

For **FFmpeg** dataset, we prepare the corresponding control flow graphs dataset as the benchmark dataset to detect binary function similarity. First, we compile *FFmpeg 4.1.4* using 2 different compilers *gcc 5.4.0* and *clang 3.8.0*, and 4 different compiler optimization levels (*O0-O3*), and generate 8 different binaries files. Second, these 8 generated binaries are disassembled using IDA Pro,[3] which can produce CFGs for all disassembled functions. Finally, for each basic block in CFGs, we extract 6 block-level numeric features as the initial node representation based on IDAPython (a python-based plugin in IDA Pro).

**OpenSSL** is built from OpenSSL (v1.0.1f and v1.0.1u) using *gcc 5.4* in 3 different architectures (x86, MIPS, and ARM), and 4 different optimization levels (*O0-O3*). The **OpenSSL** dataset we evaluate is previously released by (Xu et al., 2017) and public available[4] with prepared 6 block-level numeric features.

Overall, for both **FFmpeg** and **OpenSSL**, each node in the control flow graphs are initialized with 6 block-level numeric features: *# of string constants*, *# of numeric constants*, *# of total instructions*, *# of transfer instructions*, *# of call instructions*, and *# of arithmetic instructions*.

#### A.1.2  REGRESSION DATASETS

Instead of directly computing the GED between two graphs $G^1$ and $G^2$, we try to learn a similarity score $s(G^1, G^2)$, which is the normalized exponential of GED in the range of $(0, 1]$. To be specific, $s(G^1, G^2) = exp^{-normGED(G^1, G^2)}, normGED(G^1, G^2) = \frac{GED(G^1, G^2)}{(|G^1| + |G^2|)/2}$, where $|G^1|$ or $|G^2|$ denotes the number of nodes of $G^1$ or $G^2$, and $normGED(G^1, G^2)$ or $GED(G^1, G^2)$ denotes the normalized/un-normalized GED between $G^1$ and $G^2$.

We employ both **AIDS700** and **LINUX1000** released by (Bai et al., 2019), which are public available.[5] Each dataset contains a set of graph pairs as well as their ground-truth GED scores, which are computed by exponential-time exact GED computation algorithm $A^*$ (Hart et al., 1968; Riesen et al., 2013). As the ground-truth GEDs of another dataset **IMDB-MULTI** is provided with in-exact approximation, we thus do not consider this dataset in our experiments.

**AIDS700** is a subset of *AIDS* dataset, a collection of AIDS antiviral screen chemical compounds from Development Therapeutics Program (DTP) in the National Cancer Institute (NCI).[6] Originally, *AIDS* dataset contains 42687 chemical compounds, where each of them can be represented as a graph with atoms as node and bonds as edges. To avoid calculating the ground-truth GED between two graphs with a large number of nodes, Bai et al. (2019) create the **AIDS700** dataset that contains 700

---

[3]IDA Pro disassembler, `https://www.hex-rays.com/products/ida/index.shtml`.

[4]`https://github.com/xiaojunxu/dnn-binary-code-similarity`.

[5]`https://github.com/yunshengb/SimGNN`.

[6]`https://wiki.nci.nih.gov/display/NCIDTPdata/AIDS+Antiviral+Screen+Data`

graphs with 10 or fewer nodes. For each graph in **AIDS700**, every node is labeled with the element type of its atom and every edge is unlabeled (i.e., bonds features are ignored).

**LINUX1000** is also a subset dataset of Linux that introduced in Wang et al. (2012). The original Linux dataset is a collection of 48747 program dependence graphs generated from Linux kernel. In this case, each graph is a static representation of data flow and control dependency within one function, with each node assigned to one statement and each edge describing the dependency between two statements. For the same reason as above that avoiding calculating the ground-truth GED between two graphs with a large number of nodes, the **LINUX1000** dataset used in Bai et al. (2019) is randomly selected and contains 1000 graphs with 10 or fewer nodes. For each graph in **LINUX1000**, both nodes and edges are unlabeled.

For both classification and regression datasets, Table 7 provides more detailed statistics.

Table 7: Summary statistics of datasets for both classification & regression tasks.

| Tasks | Datasets | Sub-datasets | # of Graphs | # of Functions | # of Nodes (Min/Max/AVG) | # of Edges (Min/Max/AVG) | AVG # of Degrees (Min/Max/AVG) |
|---|---|---|---|---|---|---|---|
| classif-ication | **FFmpeg** | [3, 200] | 83008 | 10376 | (3/200/18.83) | (2/332/27.02) | (1.25/4.33/2.59) |
| | | [20, 200] | 31696 | 7668 | (20/200/51.02) | (20/352/75.88) | (1.90/4.33/2.94) |
| | | [50, 200] | 10824 | 3178 | (50/200/90.93) | (52/352/136.83) | (2.00/4.33/3.00) |
| | **OpenSSL** | [3, 200] | 73953 | 4249 | (3/200/15.73) | (1/376/21.97) | (0.12/3.95/2.44) |
| | | [20, 200] | 15800 | 1073 | (20/200/44.89) | (2/376/67.15) | (0.12/3.95/2.95) |
| | | [50, 200] | 4308 | 338 | (50/200/83.68) | (52/376/127.75) | (2.00/3.95/3.04) |
| regre-ssion | **AIDS700** | - | 700 | - | (2/10/8.90) | (1/14/8.80) | (1.00/2.80/1.96) |
| | **LINUX1000** | - | 1000 | - | (4/10/7.58) | (3/13/6.94) | (1.50/2.60/1.81) |

## A.2 MORE EXPERIMENTAL SETUP FOR MODELS

### A.2.1 MORE EXPERIMENTAL SETUP FOR BASELINE METHODS

We modified some experimental settings in baselines to fit specific tasks. Detailed settings are given in the following.

**SimGNN**: For regression task, we set batch size to 128 and trained the model for 10000 iterations. We used MSE loss as the training loss and set learning rate to $1.0 \times 10^{-3}$. Validation starts at the 9000-th iteration and is performed every 50 iterations. The best model among all the validation runs is used for the testing phase. As for the model structure, we used 3 GCNs in the first place to propagate node features, whose output dimension is set to 64, 64, and 32 respectively. Then we applied an ANPM layer implemented by author of SimGNN to perform graph-level interaction. The pair feature vector generated by this layer then passes 4 fully-connected layers and finally a 1-D scalar, which is the predicted similarity score. For classification task, we modified the training settings so now the model is trained in epoches with the same learning rate and the batch size becomes 5. Validation is carried out every iteration and the best model is saved at this time. For training loss, we used Cross-Entropy loss and applied the same learning rate as that in regression task. The model structure is the same as that in regression task except that the final output dimension of fully-connected layers is 2, the softmax of which is the predicted label and is compared with ground truth label in one-hot encoding to get the Cross-Entropy loss.

**GMN**: For classification task, we trained the model for 100 epoches where validation is carried out per epoch. We set batch size to 10 and learning rate to $1.0 \times 10^{-3}$. We set node feature dimension to 32 and graph representation dimension to 128. As for mode structure, we used 1-layer MLP as node encoder which encodes the initial node features to node states $h_i^{(0)}$. Then there are 5 propagation layers with the same structure as mentioned in the original paper (Li et al. (2019)). Besides, in this task we applied the pairwise loss based on hamming similarity defined by the author in paper. For regression task, we concatenated representation vectors of 2 graphs in the pair and passed it to a 4-layer MLP to get similarity score. As for training, we set batch size to 128 and used the same training strategy as that in the regression task of SimGNN. MSE loss is applied in this task for learning. Other settings remain the same as in classification task.

### A.2.2 More experimental setup for different GNNs

When performing experiments to see how different GNNs affect performance of MPNGMN model, we only replace GCN with GraphSAGE, GIN, and GGNN using the geometric deep learning library - PyTorch Geometric[7]. More specifically, for GraphSAGE, we used a 3-layer GraphSAGE GNN with their output dimensions all set to 100. For GIN, we used 3 GIN modules with a 1-layer MLP with output dimension 100 as the learnable function. For GGNN, we used 3 one-layer propagation models to replace the 3 GCNs in our original setting and also set their output dimensions to 100.

### A.3 More experiments of the SGNN model with different aggregation functions for both classification & regression tasks

To further compare our models with SGNN models, we train and evaluate several SGNN models with different aggregation functions, such as Max, FCMax, Avg, FCAvg, and BiLSTM. Both classification results and regression results are summarized in Table 8 and Table 9. For both classification and regression tasks, our models show statistically significantly improvement over all SGNN models with different aggregation functions, which indicates the advantage of multi-perspective node-graph matching network that adopted in our model.

Table 8: Classification results of SGNN models with different aggregation functions VS. our MP-NGMN and HGMN models in term of AUC scores (%).

| Model | FFmpeg | | | OpenSSL | | |
|---|---|---|---|---|---|---|
| | [3, 200] | [20, 200] | [50, 200] | [3, 200] | [20, 200] | [50, 200] |
| SGNN (BiLSTM) | 96.92±0.13 | 97.62±0.13 | 96.35±0.33 | 95.24±0.06 | 96.30±0.27 | 93.99±0.62 |
| SGNN (Max) | 93.92±0.07 | 93.82±0.28 | 85.15±1.39 | 91.07±0.10 | 88.94±0.47 | 82.10±0.51 |
| SGNN (FCMax) | 95.37±0.04 | 96.29±0.14 | 95.98±0.32 | 92.64±0.15 | 93.79±0.17 | 93.21±0.82 |
| SGNN (Avg) | 95.61±0.05 | 96.09±0.05 | 96.70±0.13 | 92.89±0.09 | 93.90±0.24 | 94.12±0.35 |
| SGNN (FCAvg) | 95.18±0.03 | 95.74±0.15 | 96.43±0.16 | 92.70±0.09 | 93.72±0.19 | 93.49±0.30 |
| MPNGMN | 97.73±0.11 | 98.29±0.21 | 96.81±0.96 | 96.56±0.12 | **97.60±0.29** | 92.89±1.31 |
| HGMN (Max) | 97.44±0.32 | 97.84±0.40 | 97.22±0.36 | 94.77±1.80 | 97.44±0.26 | 94.06±1.60 |
| HGMN (FCMax) | **98.07±0.06** | **98.29±0.10** | **97.83±0.11** | 96.87±0.24 | 97.59±0.24 | 95.58±1.13 |
| HGMN (BiLSTM) | 97.56±0.38 | 98.12±0.04 | 97.16±0.53 | **96.90±0.10** | 97.31±1.07 | **95.87±0.88** |

Table 9: Results of SGNN models with different aggregation functions VS. our MPNGMN and HGMN models on **AIDS700** and **LINUX1000**.

| Datasets | Model | $mse$ $(10^{-3})$ | $\rho$ | $\tau$ | p@10 | p@20 |
|---|---|---|---|---|---|---|
| AIDS700 | SGNN (BiLSTM) | 1.422±0.044 | 0.881±0.005 | 0.718±0.006 | 0.376±0.020 | 0.472±0.014 |
| | SGNN (Max) | 2.822±0.149 | 0.765±0.005 | 0.588±0.004 | 0.289±0.016 | 0.373±0.012 |
| | SGNN (FCMax) | 3.114±0.114 | 0.735±0.009 | 0.554±0.008 | 0.278±0.021 | 0.364±0.017 |
| | SGNN (Avg) | 1.453±0.015 | 0.876±0.002 | 0.712±0.002 | 0.353±0.007 | 0.444±0.012 |
| | SGNN (FCAvg) | 1.658±0.067 | 0.857±0.007 | 0.689±0.008 | 0.305±0.018 | 0.399±0.021 |
| | MPNGMN | 1.191±0.048 | 0.904±0.003 | 0.749±0.005 | **0.465±0.011** | 0.538±0.007 |
| | HGMN (Max) | 1.210±0.020 | 0.900±0.002 | 0.743±0.003 | 0.461±0.012 | 0.534±0.009 |
| | HGMN (FCMax) | 1.205±0.039 | 0.904±0.002 | 0.749±0.003 | 0.457±0.014 | 0.532±0.016 |
| | HGMN (BiLSTM) | **1.169±0.036** | **0.905±0.002** | **0.751±0.003** | 0.456±0.019 | **0.539±0.018** |
| LINUX 1000 | SGNN (BiLSTM) | 2.140±1.668 | 0.935±0.050 | 0.825±0.100 | 0.978±0.012 | 0.965±0.007 |
| | SGNN (Max) | 11.832±0.698 | 0.566±0.022 | 0.404±0.017 | 0.226±0.106 | 0.492±0.190 |
| | SGNN (FCMax) | 17.795±0.406 | 0.362±0.021 | 0.252±0.015 | 0.239±0.000 | 0.241±0.000 |
| | SGNN (Avg) | 2.343±0.453 | 0.933±0.012 | 0.790±0.017 | 0.778±0.048 | 0.811±0.050 |
| | SGNN (FCAvg) | 3.211±0.318 | 0.909±0.004 | 0.757±0.008 | 0.831±0.163 | 0.813±0.159 |
| | MPNGMN | 1.561±0.020 | 0.945±0.002 | 0.814±0.003 | 0.743±0.085 | 0.741±0.086 |
| | HGMN (Max) | 1.054±0.086 | 0.962±0.003 | 0.850±0.008 | 0.877±0.054 | 0.883±0.047 |
| | HGMN (FCMax) | 1.575±0.627 | 0.946±0.019 | 0.817±0.034 | 0.807±0.117 | 0.784±0.108 |
| | HGMN (BiLSTM) | **0.439±0.143** | **0.985±0.005** | **0.919±0.016** | **0.955±0.011** | **0.943±0.014** |

---

[7]https://pytorch-geometric.readthedocs.io/.

### A.4   More experiments of the MPNGMN model with different aggregation functions for both classification & regression tasks

We investigate the impact of different aggregation functions adopted by Aggregation Layers of the MPNGMN model for both classification and regression tasks. Following the default and same settings of previous experiments, we only change the aggregation layer of MPNGMN and use five possible aggregation functions: Max, FCMax, Avg, FCAvg, LSTM, and BiLSTM. As can be observed from Table 10 and Table 11, BiLSTM offers superior performance over all datasets of both classification and regression tasks in terms of most evaluation metrics. Therefore, we take BiLSTM as the default aggregation function for MPNGMN, and fix it for the MPNGMN part in HGMN models.

Table 10: Classification results of MPNGMN models with different aggregation functions in term of AUC scores (%).

| Model | FFmpeg | | | OpenSSL | | |
|---|---|---|---|---|---|---|
| | [3, 200] | [20, 200] | [50, 200] | [3, 200] | [20, 200] | [50, 200] |
| MPNGMN (Max) | 73.74±8.30 | 73.85±1.76 | 77.72±2.07 | 67.14±2.70 | 63.31±3.29 | 63.02±2.77 |
| MPNGMN (FCMax) | 97.28±0.08 | 96.61±0.17 | 96.65±0.30 | 95.37±0.19 | 96.08±0.48 | **95.90±0.73** |
| MPNGMN (Avg) | 85.92±1.07 | 83.29±4.49 | 85.52±1.42 | 80.10±4.59 | 70.81±3.41 | 66.94±4.33 |
| MPNGMN (FCAvg) | 95.93±0.21 | 73.90±0.70 | 94.22±0.06 | 93.38±0.80 | 94.52±1.16 | 94.71±0.86 |
| MPNGMN (LSTM) | 97.16±0.42 | 97.02±0.99 | 84.65±6.73 | 96.30±0.69 | 97.51±0.82 | 89.41±8.40 |
| MPNGMN (BiLSTM) | **97.73±0.11** | **98.29±0.21** | **96.81±0.96** | **96.56±0.12** | **97.60±0.29** | 92.89±1.31 |

Table 11: Regression results of MPNGMN models with different aggregation functions on **AIDS700** and **LINUX1000**.

| Datasets | Model | $mse$ $(10^{-3})$ | $\rho$ | $\tau$ | p@10 | p@20 |
|---|---|---|---|---|---|---|
| **AIDS 700** | MPNGMN (Max) | 2.378±0.244 | 0.813±0.015 | 0.642±0.013 | **0.578±0.199** | **0.583±0.169** |
| | MPNGMN (FCMax) | 2.220±1.547 | 0.808±0.145 | 0.656±0.122 | 0.425±0.078 | 0.504±0.064 |
| | MPNGMN (Avg) | 1.524±0.161 | 0.880±0.010 | 0.717±0.012 | 0.408±0.044 | 0.474±0.027 |
| | MPNGMN (FCAvg) | 1.281±0.075 | 0.895±0.006 | 0.737±0.008 | 0.453±0.015 | 0.527±0.016 |
| | MPNGMN (LSTM) | 1.290±0.037 | 0.895±0.004 | 0.737±0.005 | 0.448±0.007 | 0.520±0.012 |
| | MPNGMN(BiLSTM) | **1.191±0.048** | **0.904±0.003** | **0.749±0.005** | 0.465±0.011 | 0.538±0.007 |
| **LINUX 1000** | MPNGMN (Max)[*] | 16.921±0.000 | - | - | - | - |
| | MPNGMN (FCMax) | 4.793±0.262 | 0.829±0.006 | 0.665±0.011 | **0.764±0.170** | **0.767±0.166** |
| | MPNGMN (Avg) | 4.050±0.594 | 0.888±0.008 | 0.719±0.012 | 0.501±0.093 | 0.536±0.112 |
| | MPNGMN (FCAvg) | 6.953±0.195 | 0.897±0.004 | 0.736±0.005 | 0.499±0.126 | 0.509±0.129 |
| | MPNGMN (LSTM) | 1.535±0.096 | 0.945±0.004 | 0.813±0.007 | 0.695±0.064 | 0.698±0.081 |
| | MPNGMN(BiLSTM) | **1.561±0.020** | **0.945±0.002** | **0.814±0.003** | 0.743±0.085 | 0.741±0.086 |

[*] As all duplicated experiments running on this setting do not converge in their training processes, their corresponding result metrics cannot be calculated.

### A.5   More experiments of the MPNGMN model with different number of perspectives for the classification task

We further investigate the impact of different number of perspectives adopted by the Node-Graph Matching Layer of the MPNGMN model for classification tasks. Following the default and same settings of previous experiments, we only change the number of perspectives (i.e., $\widetilde{d} = 50/75/100/125/150$) of MPNGMN. As shown in Figure 2 and Table 12, when the graph size is [3, 200] and [20, 200] (more training samples), our model performance is not sensitive to the number of perspectives (from 50 to 150). When the graph size is [50,200] (fewer training samples), the variance of the model becomes relatively larger than these on [3, 200] and [20, 200]. However, when we used more perspectives (like 150), the variance of the model reduced significantly.

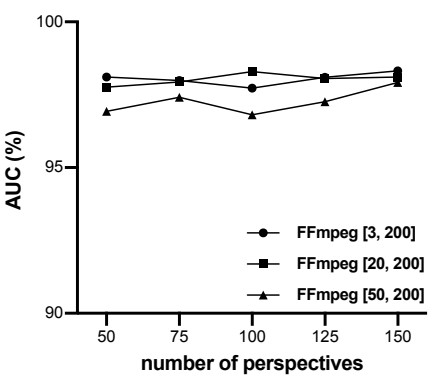
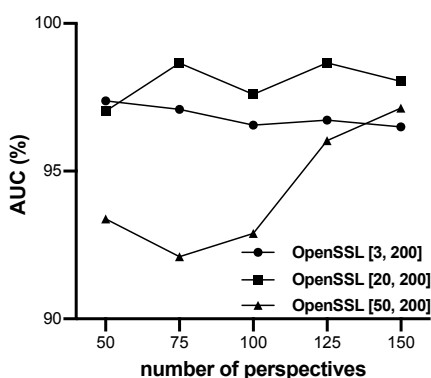

(a) classification performance of FFmpeg  (b) classification performance of OpenSSL

Figure 2: (a) and (b) show the impact of the number of perspectives on classification performance of FFmpeg and OpenSSL respectively.

Table 12: Classification results of different number of perspectives in terms of AUC scores(%).

| Model | FFmpeg | | | OpenSSL | | |
|---|---|---|---|---|---|---|
| | [3, 200] | [20, 200] | [50, 200] | [3, 200] | [20, 200] | [50, 200] |
| MPNGMN ($\tilde{d} = 50$) | 98.11±0.14 | 97.76±0.14 | 96.93±0.52 | **97.38±0.11** | 97.03±0.84 | 93.38±3.03 |
| MPNGMN ($\tilde{d} = 75$) | 97.99±0.09 | 97.94±0.14 | 97.41±0.05 | 97.09±0.25 | 98.66±0.11 | 92.10±4.37 |
| MPNGMN ($\tilde{d} = 100$) | 97.73±0.11 | **98.29±0.21** | 96.81±0.96 | 96.56±0.12 | 97.60±0.29 | 92.89±1.31 |
| MPNGMN ($\tilde{d} = 125$) | 98.10±0.03 | 98.06±0.08 | 97.26±0.36 | 96.73±0.33 | **98.67±0.11** | 96.03±2.08 |
| MPNGMN ($\tilde{d} = 150$) | **98.32±0.05** | 98.11±0.07 | **97.92±0.09** | 96.50±0.31 | 98.04±0.03 | **97.13±0.36** |

