# OpenReview forum: "Hierarchical Graph Matching Networks for Deep Graph Similarity Learning"
_ICLR.cc/2020/Conference — Reject_

### Official Review · AnonReviewer1 · 2019-10-21
**Official Blind Review #1**

**Rating:** 6

**Review:**

The paper proposes a network learning similarity between graphs. In particular, the proposed method focuses on node-graph cross-level interaction which has not been considered by other neural net studies. The performance is evaluated by two datasets on classification and regression tasks respectively.

Overall, the basic idea would be reasonable, and the architecture is clearly described. Any of my comments below are not critical concerns.

The network considers node-graph interaction, but in general, subgraph-graph interaction can be considered. Rationale that only focusing on node-graph interaction is not mentioned.

The authors repeatedly mention that the proposed method jointly learns representation and similarity. Is the learned representation can be used for any purpose? Since it depends on a counterpart of the input pair, the representation is seemingly difficult to use.

The experiments show superior performance of the proposed methods, but the datasets are only two for each tasks. In particular, since graph classification is a popular task, evaluation on a variety of benchmarks would be more convincing.

A baseline with some graph kernel can be informative.

Showing an example of graph pairs, in which cross-level interaction is indispensable to appropriately evaluate similarity, would be convincing.

A related paper being missed would be
'Yoshida, et al. Learning Interpretable Metric between Graphs: Convex Formulation and Computation with Graph Mining, SIGKDD 2019'.

**Experience Assessment:**

I have read many papers in this area.

**Review Assessment: Checking Correctness Of Derivations And Theory:**

I did not assess the derivations or theory.

**Review Assessment: Checking Correctness Of Experiments:**

I assessed the sensibility of the experiments.

**Review Assessment: Thoroughness In Paper Reading:**

I read the paper at least twice and used my best judgement in assessing the paper.

---

> ### Author Response · Authors · 2019-11-12
> **Responses for Official Blind Review #1**
>
> Author response:
>
> First of all, we want to thank the reviewer for their thorough reading and valuable comments! However, there are some points of misunderstanding that we address in this rebuttal.
>
> Below we address the concerns mentioned in the review:
>
> 1) The paper proposes a network learning similarity between graphs. In particular, the proposed method focuses on node-graph cross-level interaction which has not been considered by other neural net studies. The performance is evaluated by two datasets on classification and regression tasks respectively. Overall, the basic idea would be reasonable, and the architecture is clearly described.
>
> We are very grateful to the reviewer for this accurate summary, and for the kind recognition of our key contributions.
>
> 2) The network considers node-graph interaction, but in general, subgraph-graph interaction can be considered. Rationale that only focusing on node-graph interaction is not mentioned.
>
> Yes, subgraph-graph interaction could be exploited as well, which we will leave it as one of the future works. However, we are not sure if the subgraph-graph interaction provides additional information beyond graph-graph interactions (global-level features) and node-graph interactions (low-level features). In this work, we are still focusing on how to develop a more effective way to capture the low-level features to augment the global-level features. This is why we mainly focused on node-graph interaction in this paper.
>
> 3) The authors repeatedly mention that the proposed method jointly learns representation and similarity. Is the learned representation can be used for any purpose? Since it depends on a counterpart of the input pair, the representation is seemingly difficult to use.
>
> This is a great question. We don’t think these learned representations for each graph are best for performing other tasks such as node classification or graph classification compared to the dedicated models for these tasks. What we tried to emphasize is that our model will learn the representations of each graph (this is always true for any DL-based data driven models) that are more suitable for computing a good similarity metric between two graphs in an end-to-end fashion.
>
> 4) The experiments show superior performance of the proposed methods, but the datasets are only two for each tasks. In particular, since graph classification is a popular task, evaluation on a variety of benchmarks would be more convincing.
>
> It seems like there are some misunderstandings regarding the datasets for graph matching. Indeed, we would like to have more datasets in order to avoid any cherry-picking results. However, it is very hard to get graph matching benchmarks mainly because it is quite hard to get the ground truth (labels). Although there are many benchmarks for graph classification task, the inputs of graph matching are different and not directly available. For instance, for graph matching, the input sample is a pair of graphs (G1, G2) and the corresponding label (Y) which essentially computes the similarity between them. For all graph classification benchmarks, they only assign each graph with a label, and do not provide any similarity information between any two graphs. One cannot simply say two graphs with the same labels can be treated as “similar”.
>
> Currently, we only have two datasets created by (Bai et al., WSDM 2019) that use graph edit distance to compute ground-truth for graph-graph regression task that are publicly available. One of the main contributions of our work is to release another two datasets (as well as these sub-datasets) that are created from the binary functions compiled from the source codes for graph-graph classification task (see more details in A.1). We hope these four datasets together can serve as good benchmarks for promoting the research in developing graph match models.
>
> 5) A baseline with some graph kernel can be informative.
>
> This is a good suggestion, although some of the datasets are quite large for kernel methods. Note that in (Li et al., ICML 2019) they compared their GMN model against the WL kernel and showed significantly better performance. Our model consistently outperforms GMN model, which may be an indirect comparison to the WL kernel.
>
> 6) Showing an example of graph pairs, in which cross-level interaction is indispensable to appropriately evaluate similarity, would be convincing.
>
> This is a great suggestion. In practice, it is not hard to find an example of graph pairs to show only cross-level interaction is able to correctly predict the label right. However, it is pretty hard to show what features (in hidden high-dimensional space) the cross-level interaction captures so that they are able to perform correctly.
>
> 7) A related paper being missed would be 'Yoshida, et al. Learning Interpretable Metric between Graphs: Convex Formulation and Computation with Graph Mining, SIGKDD 2019'.
>
> Thanks for pointing it out. We have fixed it.

---

### Official Review · AnonReviewer3 · 2019-10-23
**Official Blind Review #3**

**Rating:** 3

**Review:**

In this paper, a hierarchical graph matching network, which considers both graph-graph interaction and node-graph interaction, is proposed. Specifically, the graph-graph interaction is modeled through graph level embeddings learned by GCN with pooling layers. While node-graph interaction is modeled using node embedding learned by GCN and attentive graph embedding aggregated from node embedding.

Some concerns about the paper are as follows:

1)	The novelty of the paper is incremental. The major contribution of the paper lies in the propose of multi-perspective matching function $f_m()$, which is somewhat similar to the Neural Tensor Networks proposed in [1] and utilized in [2] Although, in [2], the Neural Tensor Network is used to measure the similarity between graph-level embeddings.
2)	Some of the technical details of the paper is not clearly presented or well explained.
a.	In Eq. (7), attentive graph-level embeddings are calculated using weighted average of its node embeddings. However, it is not clear which node $i$ from the other graph should be used to calculate the weights (\alpha_{I,j}, \beta_{i,j}). Furthermore, it is also not clear why the attention score should solely base on a single node from the other graph rather than the information of the entire graph.
b.	In Eq. (10), it would be better if the authors could provide more motivations about using Bi-LSTM aggregator. Especially, the embeddings to be aggregated are unordered. What are the two directions in Bi-LSTM in this case? What is the benefit of using Bi-LSTM as aggregator compared with LSTM aggregator or other aggregators?

Some other questions to be clarified:
1)	Why different similarity score functions are adopted for the classification task and the regression task?
2)	For the classification task, the mean squared error loss is adopted. Why not using other more commonly used loss for classification task?

Suggestions:

It would be better if the authors could empirically show the effectiveness of the Bi-LSTM aggregator.

It would be helpful if the authors could conduct some investigation on how the number of perspectives affect the performance of the model.

[1] Reasoning With Neural Tensor Networks for Knowledge Base Completion
[2] SimGNN: A Neural Network Approach to Fast Graph Similarity Computation





**Experience Assessment:**

I have published one or two papers in this area.

**Review Assessment: Checking Correctness Of Derivations And Theory:**

I carefully checked the derivations and theory.

**Review Assessment: Checking Correctness Of Experiments:**

I assessed the sensibility of the experiments.

**Review Assessment: Thoroughness In Paper Reading:**

I read the paper thoroughly.

---

> ### Author Response · Authors · 2019-11-12
> **Responses for Official Blind Review #3**
>
> Author response:
>
> First of all, we want to thank the reviewer for their thorough reading and valuable comments! However, there are some points of misunderstanding that we address in this rebuttal.
>
> Below we address the concerns mentioned in the review:
>
> 1) The novelty of the paper is incremental. The major contribution of the paper lies in the propose of multi-perspective matching function , which is somewhat similar to the Neural Tensor Networks proposed in [1] and utilized in [2] ...
>
> This work has two main contributions. First of all, we proposed a new type of interactions - cross-level node-graph interactions in order to more effectively exploit different-level granularity features between a pair of input graphs, where previous works only considered graph-graph and node-node interactions. Secondly, we have provided systematic studies on different factors on the performance of all graph matching models such as the impact of different tasks (classification and regression) and the sizes of input graphs. These tasks and factors the previous model failed to fully consider them are most important aspects to validate a graph matching model is indeed better or not.
>
> It seems like there are some misunderstandings between our multi-perspective matching function and neural tensor networks. Given two input vectors v1, and v2 \in R{d}, the neural tensor networks essentially performs the following calculations:
>          h = f( v1, v2 )  =  f( v1 * T^[1,..,k] * v2), where T^[1, …,k] \in R^{d, d, k} is a tensor
> While our multi-perspective matching function performs this operation:
>       h = f( v1, v2 )  =  f( v1 .* w_i,  v2 .* w_i), where w_i \in R^{d}  is a weight vector, i =1, …,k
> And the operator .* is the element-wise multiplication.
> Therefore, we can clearly see that these two functions are very different. Conceptually, our multi-perspective matching function belongs to the class of multi-head attention function in (Vaswani et al., 2017) although it is also different. We also performed the comparisons between these two attention functions in Sec 4.4 and showed that our multi-perspective matching function is much more effective.
>
> 2a) In Eq. (7), attentive graph-level embeddings are calculated using weighted average of its node embeddings. However, it is not clear which node from the other graph should be used to calculate the weights (\alpha_{I,j}, \beta_{i,j}). Furthermore, it is also not clear why the attention score should solely base on a single node from the other graph rather than the information of the entire graph.
>
> The weights are defined and calculated in Equation (6), where we calculate the cross-graph attention coefficients between the node v_i from graph G_1 or G_2 and all other nodes v_j from other graph. In other words, each node v_i in G_1 will compute the attention coefficients (\alpha_{i,j}) between itself and all other nodes in another graph G_2. Similarly,  each node v_i in G_2 will also compute the attention coefficients (\beta_{i,j}) between itself and all other nodes in another graph G_1. We have updated our manuscript to make it more clear.
>
> 2b) In Eq. (10), it would be better if the authors could provide more motivations about using Bi-LSTM aggregator. Especially, the embeddings to be aggregated are unordered. What are the two directions in Bi-LSTM in this case? What is the benefit of using Bi-LSTM as aggregator compared with LSTM aggregator or other aggregators?
>
> This is a great question. To aggregate these cross-level interaction feature matrix from the node-graph matching layer, we employ the BiLSTM model to aggregate the unordered feature embeddings \widetilde{\vec{h}}_j. You are absolutely right that the Bi-LSTM is a biased aggregator and we just simply used any random order to perform this operation. The reason we chose this option is because BiLSTM aggregator achieved consistently (slightly) better performance compared to other aggregators as shown in our experiments. We have also observed similar choices in the previous works (Hamilton et al., NIPS 2017; Zhang et al., KDD 2019).
>
> Models	         | FFmpeg [20, 200]| FFmpeg [50, 200] | OpenSSL[3, 200]| OpenSSL[20, 200]| OpenSSL [50, 200]
> MPNGMN (Max)      	| 73.85+/-1.76 | 77.72+/-2.07 | 67.14+/-2.70 | 63.31+/-3.29 | 63.02+/-2.77
> MPNGMN (FCMax)      | 96.61+/-0.17 | 96.65+/-0.30 | 95.37+/-0.19 | 96.08+/-0.48 | 95.90+/-0.73
> MPNGMN (Avg)      	| 83.29+/-4.49 | 85.52+/-1.42 | 80.10+/-4.59 | 70.81+/-3.41 | 66.94+/-4.33
> MPNGMN (FCAvg)       | 73.90+/-0.70 | 94.22+/-0.06 | 93.38+/-0.80 | 94.52+/-1.16 | 94.71+/-0.86
> MPNGMN (LSTM)     	| 97.02+/-0.99 | 84.65+/-6.73 | 96.30+/-0.69 | 97.51+/-0.82 | 89.41+/-8.40
> MPNGMN (BiLSTM)     | 98.29+/-0.21 | 96.81+/-0.96 | 96.56+/-0.12 | 97.60+/-0.29 | 92.89+/-1.31
>
> In order to justify this choice, we have added these additional experimental results in Table 9 and 10  in the appendix in the updated manuscript.

---

> > ### Author Response · Authors · 2019-11-12
> > **Responses for Official Blind Review #3 (Continued)**
> >
> > 3) Why different similarity score functions are adopted for the classification task and the regression task?
> >
> > This is simply because two different tasks have different requirements. For the classification task, we chose cosine similarity (between -1 and 1) because we want to calculate the similarity between two graphs. For a pair of inputs, it is common to choose cosine similarity for performing classification task. For the regression task, we cannot directly use cosine similarity because the regression score range should be between 0 and 1. Therefore, we chose sigmoid as the activation function to enforce the final score range between 0 and 1.
> >
> > 4) For the classification task, the mean squared error loss is adopted. Why not using other more commonly used loss for classification task?
> >
> > The reason we chose the mean squared loss instead of other commonly used loss like cross-entropy loss is because what we really care about is still the calculation of the similarity between two graphs, instead of performing binary classification. For instance, we can train our model for classification, but we used it for graph retrieval task where we can use similarity score for ranking. Note that we used AUC score rather than accuracy as our metric.
> >
> > 5) It would be better if the authors could empirically show the effectiveness of the Bi-LSTM aggregator.
> >
> > Please see our responses above.
> >
> > 6) It would be helpful if the authors could conduct some investigation on how the number of perspectives affect the performance of the model.
> >
> > This is a great suggestion. Based on the reviewer’s advice, we have performed additional experiments to show the effect of the number of perspectives on the model performance. As shown in the following table, when the graph size is [3, 200] and [20, 200] (more training samples), our model performance is not sensitive to the number of perspectives (from 50 to 150). When the graph size is [50,200] (fewer training samples), the variance of the model becomes relatively larger than these on [3, 200] and [20, 200]. However, when we used more perspectives (like 150), the variance of the model reduced significantly.
> >
> > Models		     | FFmpeg[3, 200] | FFmpeg [20, 200] | FFmpeg [50, 200] | OpenSSL[3, 200] | OpenSSL[20, 200] | OpenSSL [50, 200]
> > MPNGMN (50)     | 98.11+/-0.14   | 97.76+/-0.14 | 96.93+/-0.52 | 97.38+/-0.11 | 97.03+/-0.84 | 93.38+/-3.03
> > MPNGMN (75)     | 97.99+/-0.09   | 97.94+/-0.14 | 97.41+/-0.05 | 97.09+/-0.25 | 98.66+/-0.11 | 92.10+/-4.37
> > MPNGMN (100)   | 97.73+/-0.11   | 98.29+/-0.21 | 96.81+/-0.96 | 96.56+/-0.12 | 97.60+/-0.29 | 92.89+/-1.31
> > MPNGMN (125)   | 98.10+/-0.03   | 98.06+/-0.08 | 97.26+/-0.36 | 96.73+/-0.33 | 98.67+/-0.11 | 96.03+/-2.08
> > MPNGMN (150)   | 98.32+/-0.05   | 98.11+/-0.07 | 97.92+/-0.09 | 96.50+/-0.31 | 98.04+/-0.03 | 97.13+/-0.36
> > Note: The number in the bracket after MPNGMN represents the number of perspectives.

---

### Official Review · AnonReviewer2 · 2019-10-28
**Official Blind Review #2**

**Rating:** 3

**Review:**

The paper proposes an architecture for (supervised) learning a similarity score between graphs through a series of layers for node embedding, node-graph matching, aggregated graph embedding, and finally prediction.

The evidence for preferring this architecture over existing one is entirely empirical -- based on experiments on four datasets.  The properties of the graphs used in the experiments are not clear: How many edges do they have on average?What is the exponent of their degree distribution on average? How many triangles do they have?  How much does the graph structure contribute to learning?   What happens if one just trained on the features of nodes?

The lack of any theoretical reasons for using this architecture over others (perhaps by linking it to the cut norm of the graphs) or insights as to when and on what types of graphs this architecture performs well reduces the significant of this paper.


**Experience Assessment:**

I have published one or two papers in this area.

**Review Assessment: Checking Correctness Of Derivations And Theory:**

I assessed the sensibility of the derivations and theory.

**Review Assessment: Checking Correctness Of Experiments:**

I carefully checked the experiments.

**Review Assessment: Thoroughness In Paper Reading:**

I read the paper thoroughly.

---

> ### Author Response · Authors · 2019-11-12
> **Responses for Official Blind Review #2**
>
> Author response:
>
> First of all, we want to thank the reviewer for providing valuable comments! Below we address the concerns mentioned in the review:
>
> 1) The evidence for preferring this architecture over existing one is entirely empirical -- based on experiments on four datasets.
>
> Our model is well motivated by the drawbacks of existing works that they either only consider graph-graph level interactions or node-node level interactions when computing the graph similarity for graph matching. As we clearly illustrated in the Introduction Section, we would like to ask how a DL model can address the following challenges in order to overcome the above limitations: : i) how to learn different-level granularity (global level and local level) of interactions between a pair of graphs; ii) how to effectively learn more rich cross-level interactions between parts of a graph and a whole graph. To effectively cope with these challenges, we propose our Hierarchical Graph Matching Network(HGMN) for computing the graph similarity between any  pair of graph-structured  objects. The proposed HGMN model consists of a novel multi-perspective node-graph matching network for effectively learning cross-level interactions between parts of a graph and a whole graph, and a siamese graph neural network for learning global-level interactions between two graphs.
>
> We would like to point it out that existing relevant works (Bai et al., WSDM 2019) and  (Li et al., ICML 2019) considered either graph-graph regressions ( AIDS700 and LINUX1000) or graph-graph classification (FFmpeg). We are the first one to systematically investigate different factors on the performance of all graph matching models such as the impact of different tasks (classification and regression on four datasets) and the sizes of input graphs (which is ignored by the existing works).
>
> 2) The properties of the graphs used in the experiments are not clear: edge, degree distribution.. How much graph structure contribute to learning?   What happens if one just trained on the features of nodes?
>
> We have listed various properties of four datasets in Table 1. Based on reviewer’s comments, we have also added the characteristics of edges in the Appendix of the updated manuscript. We copied some of them below.
>
> Datasets 		     |  # of Edges (Min/Max/AVG)	| AVG # of Degrees (Min/Max/AVG)
> FFmpeg [3, 200]	     | (2/332/27.02) 		                | (1.25/4.33/2.59)
> FFmpeg [20, 200]	     | (20/352/75.88) 		                | (1.90/4.33/2.94)
> FFmpeg [50, 200]	     | (52/352/136.83) 		                | (2.00/4.33/3.00)
> OpenSSL [3, 200]	     |  (1/376/21.97) 		                | (0.12/3.95/2.44)
> OpenSSL [20, 200]    |  (2/376/67.15)		                | (0.12/3.95/2.95)
> OpenSSL [50, 200]    |  (52/376/127.75)		                | (2.00/3.95/3.04)
> AIDS700		            |  (1/14/8.80) 			                | (1.00/2.80/1.96)
> LINUX1000		    |  (3/13/6.94) 			                | (1.50/2.60/1.81)
> -----------------------------------------------------------------------------------------------
>
> GNN takes both graph structure and node features as inputs. As we can see in Table 1, datasets AIDS700, FFmpeg, and OpenSSL have rich node features while dataset LINUX1000 only has one node feature. In our extensive experiments, our model is able to consistently outperform other state-of-the-art baselines on these datasets with quite different characteristics, highlighting that our model is not sensitive to the graph structure and the node features.
>
> We are not sure what the reviewer mean “What happens if one just trained on the features of nodes?”. All GNN models must have graph structure (graph adjacency matrix) as inputs and cannot be trained only on node features.
>
> (3) The lack of any theoretical reasons ... or insights as to when and on what types of graphs this architecture performs well ...
>
> This is a good suggestion and we will leave it as one of our future works. However, all previous works (Bai et al., WSDM 2019) and  (Li et al., ICML 2019) did not provide any theories to support their models either.
>
> For graph matching task, our model and two previous models  (Bai et al., WSDM 2019) and  (Li et al., ICML 2019) did not assume the types of graphs to consider. However, it is important to have a model that considers both global-level interactions and low-level interactions between two graphs. This is exactly our main contribution in this work that we proposed a novel type of interactions - cross-level node-graph interactions in order to more effectively exploit different-level granularity features between a pair of input graphs. In addition, we have provided systematic studies on different factors on the performance of all graph matching models such as the impact of different tasks (classification and regression) and the sizes of input graphs. These tasks and factors the previous model failed to fully consider them are most important aspects to validate a graph matching model is indeed better or not.

---

### Decision · Program_Chairs · 2019-12-19

**Decision:**

Reject

**Comment:**

The submission proposes an architecture to learn a similarity metric for graph matching. The architecture uses node-graph information in order to learn a more expressive, multi-level similarity score. The hierarchical approach is empirically validated on a limited set of graphs for which pairwise matching information is available and is shown to outperform other methods for classification and regression tasks.

The reviewers were divided in their scores for this paper, but all noted that the approach was somewhat incremental and empirically motivated, without adequate analysis, theoretical justification, or extensive benchmark validation.

Although the approach has value, more work is needed to support the method fully. Recommendation is to reject at this time.